# Do Serum Galectin-9 Levels in Women with Gestational Diabetes and Healthy Ones Differ before or after Delivery? A Pilot Study

**DOI:** 10.3390/biom13040697

**Published:** 2023-04-20

**Authors:** Aleksandra Pełech, Monika Ruszała, Magdalena Niebrzydowska-Tatus, Katarzyna Bień, Żaneta Kimber-Trojnar, Monika Czuba, Małgorzata Świstowska, Bożena Leszczyńska-Gorzelak

**Affiliations:** 1Department of Obstetrics and Perinatology, Medical University of Lublin, 20-090 Lublin, Poland; monika.ruszala@wp.pl (M.R.); mniebrzydowska7@gmail.com (M.N.-T.); kaasiabien@gmail.com (K.B.); bozena.leszczynska-gorzelak@umlub.pl (B.L.-G.); 2Department of Clinical Genetics, Medical University of Lublin, 20-080 Lublin, Poland; monika.czuba@umlub.pl (M.C.); malgorzata.swistowska@umlub.pl (M.Ś.)

**Keywords:** galectin-9, gestational diabetes mellitus, bioelectrical impedance analysis, pregnancy

## Abstract

Gestational diabetes mellitus (GDM) is a common metabolic disease that occurs during pregnancy, with the placenta playing an important role in its pathophysiology. Currently, the role of galectin-9 in the development of GDM is unknown. The aim of this study was to compare galectin-9 concentrations in healthy pregnant women and those with GDM. Galectin-9 levels were assessed in serum samples taken both just before and after delivery, as well as in urine samples collected in the postpartum period. Maternal body composition and hydration status were evaluated using the bioelectrical impedance analysis (BIA) method. There were no statistically significant differences in the concentration of galectin-9 in women with GDM compared to healthy pregnant women in their serum samples taken just before delivery, nor in their serum and urine samples collected in the early postpartum period. However, serum galectin-9 concentrations taken before delivery were positively correlated with BMI and parameters related to the amount of adipose tissue assessed in the early postpartum period. Additionally, there was a correlation between serum galectin-9 concentrations taken before and after delivery. Galectin-9 is unlikely to become a diagnostic marker for GDM. However, this subject requires further clinical research in larger groups.

## 1. Introduction

Gestational diabetes mellitus (GDM) is the most common metabolic disease that complicates pregnancy [1,2]. GDM is classified as one of the “great obstetric syndromes,” in which the role of the placenta in the interactions of the maternal and fetal unit is crucial [3]. Its prevalence ranges from 6% to 18% of all pregnancies and still seems to be increasing [4]. The condition is associated with numerous health issues affecting pregnant women, such as an increased incidence of gestational hypertension, preeclampsia, and urinary tract infections [5,6,7]. Persistently elevated maternal blood glucose also affects the developing fetus, causing excessive production of fetal insulin, resulting in a higher incidence of macrosomia and birth trauma, especially shoulder dystocia [8,9]. In order to avoid GDM-related complications, early detection and effective treatment are essential. It is also important to remember that both the mother and offspring are not only predisposed to developing type 2 diabetes mellitus (T2DM) in the future, but also to adiposity, cardiovascular diseases, and metabolic syndrome [1,10,11].

Pregnancy is associated with a physiological decrease in the insulin sensitivity of tissues [12,13]. While most women manage to maintain euglycemia, some will have their compensatory mechanisms disrupted and develop impaired glucose tolerance [5]. The pathogenesis of GDM is still not fully known, but the mechanisms of its development seem to be similar to T2DM. Moreover, the risk factors in both conditions are comparable, such as obesity, physical inactivity, and genetic predisposition [1,14,15]. The development of T2DM is associated with low-grade inflammation, causing an increased concentration of pro-inflammatory cytokines, which can interfere with the insulin response mechanisms. This causes an increase in insulin production and secretion, and eventually insulin resistance [16].

Galectins are a group of 16 carbohydrate-binding proteins with a resemblance to β-galactosides [17]. They are expressed in the intestinal tract, endothelial cells, and also in the endometrium and placenta [14]. Their functions affect cell growth, differentiation, and apoptosis, suggesting their link to carcinogenesis [4,14,18]. It has been observed that galectin concentrations can rise significantly as a result of increased production by cancer cells, but also as a consequence of stress conditions such as cell damage and inflammation. Due to such qualities, these molecules could find use as potential biomarkers for metabolic diseases and cancers [17].

Moreover, galectins seem to have a significant role in immunomodulatory processes in early pregnancy, during trophoblast invasion and angiogenesis, suggesting that they are crucial for maintaining a healthy pregnancy [19]. Researchers are looking for a link between galectins and pregnancy complications such as preeclampsia, intrauterine growth restriction, HELLP syndrome, and the premature rupture of fetal membranes [4,19,20], but it is still unclear. According to the research, galectin-13 levels in the first trimester of pregnancy, also known as placental protein-13 (PP-13), seem to correlate with the occurrence of preeclampsia [14].

Associations between galectins and the occurrence of GDM are also being explored. Other studies have demonstrated a correlation between elevated levels of PP-13 in the second trimester of pregnancy and the development of GDM [21]. Other research aimed to evaluate the association of galectin-4, an immunomodulator of proinflammatory functions, with the pathophysiology of GDM. As a result, it demonstrated the overexpression of galectin-4 in the decidua of women diagnosed with GDM [22]. There is evidence that elevated levels of circulating galectin-3 are correlated with the presence of GDM [4,8,11]. According to the research, through its effect on the development of insulin resistance, it mediates the pathogenesis of GDM [4]. Moreover, it is reported that galectin-3 levels are reduced in cord blood, which may suggest its potential fetal-damaging effects, as well as proving the protective role of the placenta, according to another study [8]. However, there are reports that galectin-3 concentrations are shown to increase over the duration of a healthy pregnancy, while also tending to be reduced in patients who developed GDM [23].

Galectin-9 is a protein expressed mainly in the liver, small intestine, and thymus [16,24]. The half-life of galectin-9 is short, ranging from 30 to 60 min. It has been discovered as a potential auto-antigen in Hodgkin’s lymphoma, possibly affecting the regulation of immune processes [17]. Galectin-9, like other galectins, plays a role in regulating cell proliferation, differentiation, and apoptosis. It binds to mucin domain-3 (Tim-3), a receptor expressed in T cells, monocytes, and natural killer (NK) cells, inducing apoptosis to de-escalate immune responses [25]. Increased galectin-9 levels have been observed in both viral and bacterial infections [26], and its association with autoimmune processes and cancer is also possible [27].

Galectin-9 appears to influence glucose homeostasis by binding to glucose transporter-2 (GLUT-2), which is expressed in pancreatic β cells [17,28]. In response to glucose stimulation, GLUT-2 mediates the entry of glucose into the cell by translocating to the plasma membrane. Galectin-9 expression has been shown to increase in subcutaneous adipose tissue and in macrophages of visceral adipose tissue in obese mice, as well as in the serum of patients with T2DM [16]. In pregnancy, trophoblast cells secrete galectin-9, which can cause Tim-3+ peripheral NK cells to acquire a decidual NK phenotype, defined by high levels of interleukin 4, low levels of tumor necrosis factor α, and reduced cytotoxic capacity. Due to its immunomodulatory properties, galectin-9 may have a significant function in fetal–maternal tolerance and pregnancy-related complications [25,29].

The role of galectin-9 in the pathogenesis of GDM is unknown, and to our knowledge, there are no publications exploring this topic. The aim of our study is to compare galectin-9 concentrations in healthy and GDM women before and after delivery, taking into account clinical parameters in both groups. Considering what is known about the functions of galectins and the pathophysiology of GDM, we hypothesized that the concentration of galectin-9 in serum would be higher in women with GDM compared to healthy controls. We also assumed that galectin-9 levels would be elevated in both groups before delivery, when the patients’ weight was higher, including placental weight, compared to its concentrations after delivery. Moreover, we also wanted to evaluate the usefulness of determining the galectin-9 levels in urine, a material that can be collected in a non-invasive way, as a potential marker of GDM.

## 2. Materials and Methods

Our study included 41 female Caucasian patients who delivered at the Chair and Department of Obstetrics and Perinatology, after completing a singleton term pregnancy (i.e., after 37 weeks of gestation). We divided the patients into two groups: the first group consisted of 18 women with GDM, while the second group consisted of 23 women with a normal singleton term pregnancy, without risk factors and without other metabolic abnormalities. All patients, both in the control group and in the GDM group, were characterized by normal pre-pregnancy body mass index (BMI; i.e., in the range of 18.5–24.9 kg/m^2^).

We selected only those patients who had a fasting plasma glucose test before 10 weeks of pregnancy of less than 92 mg/dL. Additionally, we followed the recommendations of Institute of Medicine (IOM), according to which the proper weight gain is between 11.5 and 16 kg.

The 75 g oral glucose tolerance test (OGTT) is the recommended diagnostic test for GDM in Poland. If the patient does not have GDM risk factors, such as obesity, no history of GDM or macrosomia, the test should be performed between 24 and 26 weeks of pregnancy, in accordance with the Regulation of the Minister of Health, which has been in force in Poland since 1 January 2019 [30].

All patients were informed about the study protocol, and a detailed written consent was obtained from each patient who agreed to participate in the study. The study protocol received approval from the Bioethics Committee of the Medical University of Lublin (KE-0254/61/2020, approved on 26 March 2020).

The recruitment rules for our study are presented in Figure 1.

We conducted a survey to gather information on the patients’ family history of diabetes or obesity and gestational weight gain. Anthropometric measurements were performed, and maternal body composition and hydration status were evaluated using the bioelectrical impedance analysis (BIA) method and a body composition monitor (BCM) (Fresenius Medical Care) in the early postpartum period (i.e., 48 h after delivery).

We measured several parameters in the serum, including glucose, insulin, hemoglobin A1c (HbA1c), uric acid, creatinine, total cholesterol, high-density lipoprotein cholesterol (HDL), low-density lipoprotein cholesterol (LDL), non-HDL, triglycerides, homocysteine, and homeostasis model assessment-insulin resistance (HOMA-IR), using a certified laboratory. We also calculated the triglyceride-glucose index (TyG) as the logarithm of the product of glucose and triglycerides, with the formula being: ln [glucose (mg/dL) × triglycerides (mg/dL)/2]. This parameter has been recommended as an alternative indicator of insulin resistance, because it correlates to glucotoxicity and lipotoxicity [31,32,33].

Galectin-9 concentrations were determined in maternal serum samples taken before delivery and in the early postpartum period (i.e., 48 h after delivery) and in urine samples taken 36–48 h after delivery using an enzyme-linked immunosorbent assay (Sandwich ELISA) and kits available on the market (R&D Systems, Inc., Minneapolis, MN, USA; Quantikine Human Galectin-9 Immunoassay; catalog number DGAL90; detection range 0.2–10 ng/mL, sensitivity 0.028 ng/mL).

We used the chi-squared test (with Yates’ correction for 2 × 2 tables) to compare qualitative variables between the groups, and Fisher’s exact test was used in cases of low values in contingency tables. The Mann–Whitney test was used to compare quantitative variables between the two groups, while the relationship between two quantitative variables was assessed using Spearman’s coefficient of correlation. We used linear regression to analyze the impact of potential predictors on quantitative variables, with regression parameters and 95% confidence intervals shown. We set the significance level for all statistical tests to 0.05, and all *p*-values below 0.05 were interpreted as indicative of significant relationships. We performed all computations using R 4.2.2.

## 3. Results

We did not observe any significant differences in fasting glycemia in the first trimester of pregnancy, gestational weight gain, BMI values (in three time frames, i.e., before pregnancy, before delivery and after delivery) and family history of diabetes and obesity between women with GDM and healthy controls.

In accordance with our principles of patient recruitment for the study, women in the GDM group had at least one abnormal result in the OGTT conducted between 24 and 26 weeks of pregnancy (Table 1).

Eighteen pregnant patients diagnosed with GDM were on a special diet. In addition, 39% of them also required one daily injection of basal insulin (the maximum daily dose of insulin was 16 units).

Among the biochemical and biophysical parameters obtained between 36 and 48 h after delivery, we found that significantly higher fasting glycemia and fat tissue index (FTI) levels were found in the group of patients with GDM compared to the healthy group of mothers (Table 2).

### 3.1. Comparison of Galectin-9 Concentration in GDM Patients and Healthy Controls

Comparing galectin-9 concentrations in both pre- and postpartum serum and urine tests, there were no statistically significant differences (*p* > 0.05) between mothers with GDM and controls (Table 3).

We found that the concentration of galectin-9 in the serum was not statistically different (*p* > 0.05) before or after delivery in the statistical calculations performed in both the healthy group and the GDM group.

### 3.2. Correlations of Galectin-9 Determinations

#### 3.2.1. Prepartum Serum Galectin-9 and Postpartum Serum Galectin-9

A significant positive relationship was found (*p* < 0.001); the higher the prepartum serum galectin-9 level, the higher the postpartum serum galectin-9 level (Figure 2).

#### 3.2.2. Prepartum Serum Galectin-9 and Postpartum Urine Galectin-9

A statistically insignificant relationship was found (*p* > 0.05).

#### 3.2.3. Postpartum Serum Galectin-9 and Postpartum Urine Galectin-9

A statistically insignificant relationship was found (*p* > 0.05).

### 3.3. Correlations with Galectin-9 Levels

The prepartum serum galectin-9 concentration correlated significantly (*p* < 0.05) and positively (r > 0) with postpartum BMI, E/I, ATM and FTI. There were no statistically significant correlations for galectin-9 levels determined postpartum in either serum or urine (Table 4).

### 3.4. Multiple Linear Regression Analyses

The multiple linear regression model showed that none of the analyzed characteristics was a significant independent predictor of the level of galectin-9 (as all *p* > 0.05) either in prepartum serum (Table 5) or postpartum serum (Table 6) and urine (Table 7).

### 3.5. Correlations with Galectin-9 Levels and the Sex of the Child

The differences were not statistically significant (*p* > 0.05)—according to calculations using the Mann–Whitney test.

### 3.6. Correlations with Galectin-9 Levels and the Mode of Delivery

The differences were not statistically significant (*p* > 0.05)—according to calculations using the Mann–Whitney test.

## 4. Discussion

Currently, the diagnosis of GDM can refer to any pregnant woman at any time during pregnancy. However, it is currently limited in the new literature only to diagnoses of impaired glucose on OGTT in the second or third trimester of pregnancy, according to the American Diabetes Association [34]. This approach therefore does not take into account diagnoses made in the first trimester of pregnancy, assuming that these women were not diagnosed with diabetes before pregnancy. For this reason, the inclusion criteria for our “GDM group” were only pregnant women with a diagnosis made at 24–26 weeks of pregnancy.

In our study, we wanted to investigate whether we would find differences in the concentration of galectin-9 before and after delivery in the same patients. Many molecules, including placental-released adipokines and cytokines, are known to lead to physiological insulin resistance. During pregnancy, with the development of the placenta, the production of hormones and proteins that act in opposition to insulin increases with each passing day. Meanwhile, after labor, due to delivery of the placenta, the concentration of placental hormones decreases rapidly, with the result that most patients return to pre-pregnancy glycemia. GDM patients are advised to return to a normal diet after giving birth.

We found that the concentration of galectin-9 in the serum was not statistically different before or after delivery in the statistical calculations performed in either the healthy group (7.8 ng/mL before delivery vs. 7.74 ng/mL after delivery) or the GDM group (9.39 ng/mL before delivery vs. 8.4 ng/mL after delivery).

Galectins are proteins that bind to β-galactosides and are involved in various biological processes, such as organogenesis, oncogenesis, cell adhesion, cell-cycle regulation, and immunity. Several molecules belonging to the galectin family have been identified as potential biomarkers for pregnancy complications, some of which have pathophysiological significance. Some of these galectins are known to play a role in the pathophysiology of GDM.

Our study focused on galectin-9, and to the best of our knowledge, it is the first study to compare galectin-9 levels in the same group of women with GDM and normal pregnancies. We assessed both serum galectin-9 levels just before and after delivery, as well as in urine during the postpartum period in patients with GDM and uncomplicated pregnancies. However, collecting urine in a sterile manner in patients just before delivery was challenging due to the obstetric situation, including the occurrence of regular uterine contractions, spotting from the genital tract due to cervical dilatation, or amniotic fluid leakage.

Galectin-9 is currently not well understood, and little is known about its exact function and properties. It is expressed in the endometrium, trophoblasts, stromal cells, and endothelial cells, and plays a role in implantation and early fetal development through local anti-inflammatory effects. The expression and concentration of galectin-9 in maternal blood have been observed to increase during pregnancy, reaching levels several times higher than in non-pregnant women. As a result, it may play a role in immune tolerance, which is crucial for fetal development [25].

Several studies have suggested a role for low galectin-9 expression and/or maternal serum levels in spontaneous miscarriages, recurrent pregnancy loss, and preeclampsia [19]. However, studies relating to galectin-9 determinations in women diagnosed with GDM have never been performed before. In our study, we compared the concentrations of galectin-9 in women with and without GDM both in prepartum serum and postpartum serum and urine. To our surprise, we did not find any statistically significant differences depending on the diagnosis, or of GDM.

Recently, galectin-9 has been shown to play an immunomodulatory role in various microbial infections. It mediates host–pathogen interactions and regulates cell signaling by binding to its receptors. It is worth noting that galectin-9 is involved in many physiological functions, including growth, differentiation, adhesion, communication, and cell death. However, recent studies emphasize the elevated levels of galectin-9 in autoimmune diseases, viral infections, cancer, acute liver failure, chronic kidney disease, coronary artery disease, arterial atherosclerosis, and gynecological-related infertility [17].

Recent studies have focused on the role of galectin-9 in the pathophysiology of T2DM. Sun et al. [27] evaluated the role of galectin-9 in the pathogenesis of obesity-related T2DM. The study was conducted in three groups—control, with simple obesity, and obesity with T2DM. The concentration of galectin-9 in the serum of obese T2DM subjects was significantly higher than in healthy individuals and in the group with simple obesity. Moreover, the level of galectin-9 in the group of obese T2DM patients was positively correlated with fasting insulin and C-peptide, which are two clinical features representing the function of the pancreatic islets in T2DM [27]. In our study, prepartum serum galectin-9 levels correlated significantly and positively with postpartum body mass index (BMI), extracellular water to intracellular water index (E/I), adipose-tissue mass (ATM) and fat-tissue index (FTI). Most of these parameters are related to the amount of adipose tissue. Our observation seems to confirm the hypothesis of Sun et al. that galectin-9 may be a potential biomarker associated with the pathogenesis of obesity-related T2DM.

Kurose et al. [35] demonstrated that the concentration of galectin-9 in the serum of patients with T2DM was significantly and negatively correlated with the estimated glomerular filtration rate (eGFR). This suggests that the increase in serum galectin-9 levels in T2DM patients is closely related to eGFR and may be associated with changes in the immune response and inflammation in patients with T2DM and chronic kidney disease. Similar results were obtained by Xie et al. [36], who indicated that serum galectin-9 levels increased with the progression of kidney damage (GFR from G1 to G4). In our study, both control and GDM patients had normal eGFR, which did not affect serum galectin-9 levels.

The association of galectin-9 with T2DM is further supported by Jia et al. [37]. They have shown that galectin-9, through its cytoplasmic AMP-activated protein kinase (AMPK) activity, can influence various health conditions affected by AMPK, including obesity, diabetes, cancer, and immune responses, and may be part of the mechanism of action of some diabetes medicines, including metformin.

In previous studies, we have confirmed that it is worth considering not being limited to the assessment of serum concentrations of the tested molecules only [38,39,40]. Urine should also be considered, as this material can provide additional information, is collected in a non-invasive way for patients, and seems to be helpful in explaining the pathogenesis of many civilization diseases, including T2DM. We found only one publication presenting the results of determining galectin-9 in urine. Mehta et al. [41] compared serum and urine galectin-9 concentrations in patients with systemic lupus erythematosus (SLE) with a control group. SLE patients had higher serum galectin-9 levels than healthy subjects, but there was no difference in urinary galectin-9 concentrations. An association was also found between the severity of SLE and the concentration of galectin-9, but only in serum. In our study, we observed no differences between healthy mothers and women with GDM in the concentrations of galectin-9 in the serum (collected before and after delivery) or in the urine (collected in the early postpartum period). However, we found correlations between serum galectin-9 concentrations before and after delivery. We observed no relationship between serum galectin-9 before delivery and urine galectin-9 after delivery, nor between serum and urine galectin-9 concentrations after delivery. Unfortunately, due to the aforementioned obstetric reasons, we could not measure urinary galectin-9 concentrations just before delivery and thus could not correlate the prepartal serum and urinary galectin-9 concentrations.

The main limitation of this study is certainly the small size of the groups. Another limitation of the study is the evaluation of galectin-9 in the urine, which was carried out in a single urine sample collected 36–48 h after delivery. We did not study the daily rate of excretion of this molecule, nor did we assess the level of creatinine in the urine. However, the results of our research may serve as an interesting starting point for other scientists.

## 5. Conclusions

Certainly, studies on galectin-9 in GDM should be conducted on a larger group of women. Based on the results, it seems worth considering testing this molecule in pregnant women with excessive weight gain. In our study, we observed that there is an association between the galectin-9 level determined before delivery and postpartum body fat results. We found no significant association between serum galectin-9 concentrations in women with GDM compared to healthy controls. These levels were also not different in either group before delivery, compared to postpartum concentrations. Furthermore, urine does not seem to be an interesting material for testing galectin-9 in pregnant women with normal renal function.

Galectin-9 is unlikely to become a diagnostic marker for GDM. However, this subject requires further clinical research in larger groups. In the future, we plan further studies to carry out follow-up with our patients and compare with the current clinical status, including anthropometric findings, development of insulin resistance and dyslipidemia.

## Figures and Tables

**Figure 1 biomolecules-13-00697-f001:**
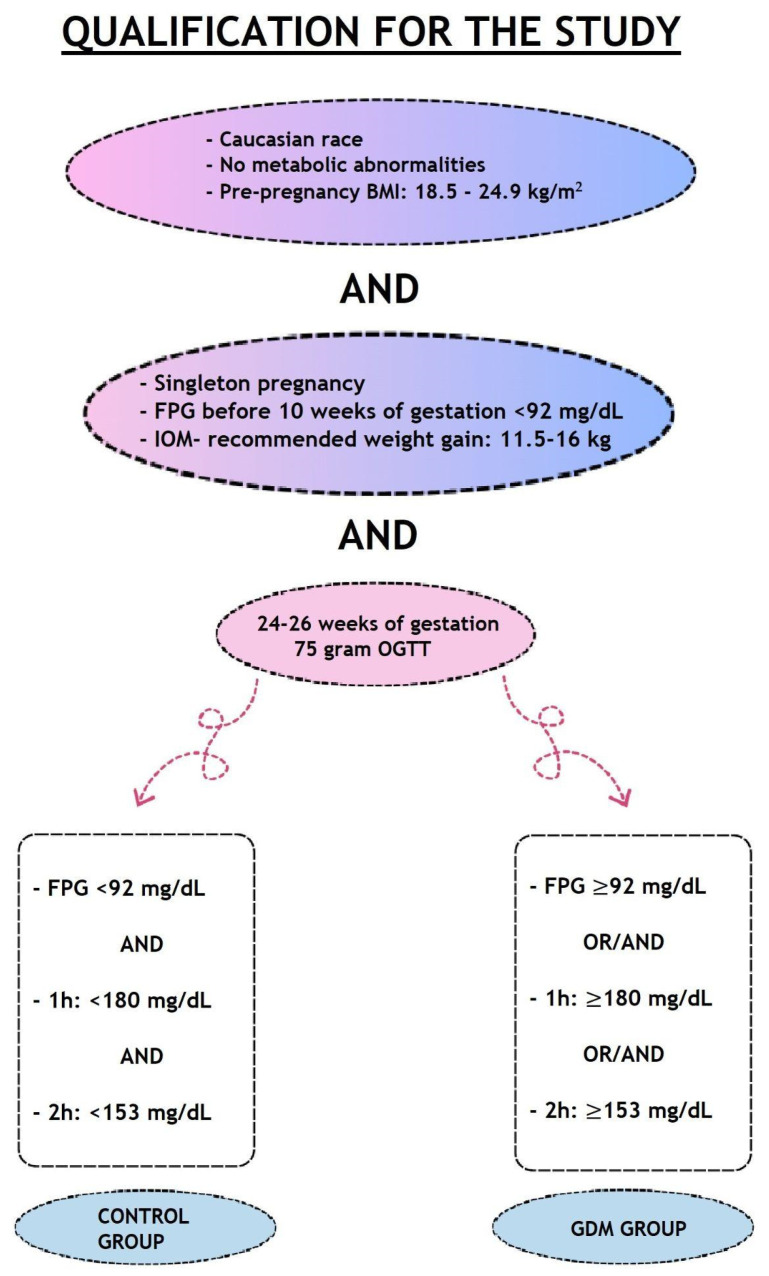
Qualification for the study.

**Figure 2 biomolecules-13-00697-f002:**
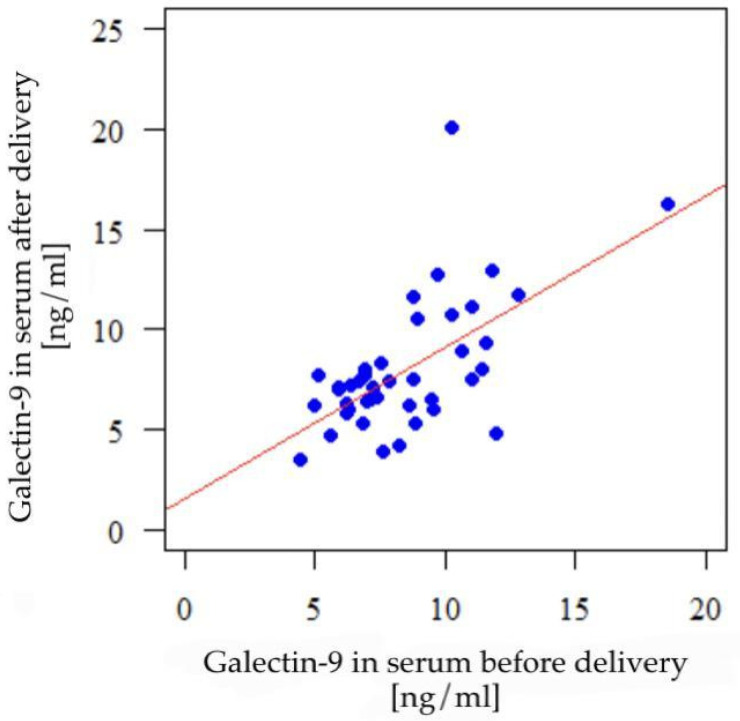
Spearman rank correlation coefficient = 0.545; *p* < 0.001.

**Table 1 biomolecules-13-00697-t001:** Results of qualification of patients with GDM.

Variables in OGTT		Glycemia
FPG (mg/dL)	Mean (SD)	100.72
	Median (quartile)	96.8
	Range	80–112
After 1 h (mg/dL)	Mean (SD)	159.84
	Median (quartile)	171
	Range	100–205
After 2 h (mg/dL)	Mean (SD)	125.86
	Median (quartile)	147
	Range	97–173

FPG—fasting plasma glucose; OGTT—the 75 g oral glucose tolerance test.

**Table 2 biomolecules-13-00697-t002:** Comparison of characteristics of the study subjects.

Variables		Control Group (n = 23)	GDM Group (n = 18)	*p*
Age (years)	Mean (SD)	31.65 (3.91)	31.94 (6.06)	*p* = 0.612
	Median (quartile)	31.62 (28.98–34.07)	32.95 (27.31–36.3)	
	Range	24.02–38.97	20.15–40.43	
Week of pregnancy termination (weeks)	Mean (SD)	39 (1.31)	38.39 (1.2)	*p* = 0.215
	Median (quartile)	39 (38–40)	38 (38–39)	
	Range	37–42	36–40	
Mode of delivery	Caesarean section	9 (39.13%)	6 (33.33%)	*p* = 0.956
	Natural labour	14 (60.87%)	12 (66.67%)	
Sex of the child	Female	6 (26.09%)	6 (33.33%)	*p* = 0.873
	Male	17 (73.91%)	12 (66.67%)	
Birth weight (g)	Mean (SD)	3451.74 (499.37)	3315.56 (397.75)	*p* = 0.312
	Median (quartile)	3440 (3040–3570)	3290 (3132.5–3432.5)	
	Range	2800–4800	2630–4480	
Parameters obtained between 36 and 48 h after delivery				
BMI (kg/m^2^)	Mean (SD)	28.04 (3.88)	31.26 (8.84)	*p* = 0.18
	Median (quartile)	27.3 (25.56–28.8)	29.75 (25.4–34.12)	
	Range	22.2–37.2	19.77–59.16	
Fasting glycemia (mg/dL)	Mean (SD)	74.48 (10.81)	84.06 (13.25)	*p* = 0.008 *
	Median (quartile)	75 (72–80.5)	83.5 (78.25–94.5)	
	Range	37–92	54–105	
Insulin (mU/L)	Mean (SD)	10.32 (4.41)	11.95 (6.7)	*p* = 0.608
	Median (quartile)	9.2 (7.65–13.9)	12 (6.32–18.4)	
	Range	3–19.6	2.1–22.7	
HbA1c (IFCC) (mmol/mol)	Mean (SD)	34.83 (3.59)	35.61 (6.34)	*p* = 0.607
	Median (quartile)	34 (32.5–36.5)	36 (30–39.5)	
	Range	30–43	25–48	
Creatinine (mg/dL)	Mean (SD)	0.54 (0.07)	0.55 (0.09)	*p* = 0.907
	Median (quartile)	0.5 (0.5–0.6)	0.5 (0.5–0.6)	
	Range	0.4–0.7	0.4–0.7	
Total cholesterol (mg/dL)	Mean (SD)	269.83 (45.88)	276.11 (50.23)	*p* = 0.646
	Median (quartile)	264 (241–299)	271.5 (241.25–315.75)	
	Range	206–387	188–354	
HDL (mg/dL)	Mean (SD)	73.43 (15.51)	77.22 (23.92)	*p* = 1
	Median (quartile)	69 (64.5–78)	72.5 (57–93.75)	
	Range	48–108	50–132	
LDL (mg/dL)	Mean (SD)	129.26 (54.24)	147.06 (34.66)	*p* = 0.379
	Median (quartile)	139 (106–160)	148 (126–176.25)	
	Range	0–222	92–207	
Triglycerides (mg/dL)	Mean (SD)	275.35 (93.08)	259 (58.99)	*p* = 0.674
	Median (quartile)	258 (214.5–328.5)	250 (214.75–304.25)	
	Range	138–517	167–353	
TyG	Mean (SD)	9.17 (0.41)	9.26 (0.33)	*p* = 0.543
	Median (quartile)	9.2 (8.95–9.4)	9.25 (9.2–9.4)	
	Range	8.2–9.9	8.6–9.8	
E/I	Mean (SD)	0.89 (0.13)	0.95 (0.2)	*p* = 0.494
	Median (quartile)	0.9 (0.86–0.95)	0.92 (0.83–0.97)	
	Range	0.59–1.15	0.61–1.4	
ATM (kg)	Mean (SD)	37.46 (22.65)	46.23 (26.59)	*p* = 0.145
	Median (quartile)	32 (26.35–40.45)	42 (30.58–60.35)	
	Range	15.4–125.6	7.3–125.6	
FTI (kg/m^2^)	Mean (SD)	12.95 (7.92)	17.11 (9.4)	***p* = 0.04 ***
	Median (quartile)	11.3 (9.4–14.25)	15.1 (12.38–21.9)	
	Range	5.5–45	2.5–45	
BCM (kg)	Mean (SD)	20.88 (6.79)	17.51 (4.49)	*p* = 0.148
	Median (quartile)	18.7 (16.6–23.85)	17.05 (14.53–19.73)	
	Range	13.9–43.5	8.6–27.3	

*p*—Qualitative variables: test chi-square or Fisher’s exact test; quantitative variables: Mann–Whitney test; *—difference statistically significant (*p* < 0.05). ATM—adipose tissue mass; BCM—body cell mass; BMI—body mass index; E/I—extracellular water to intracellular water index; FTI—fat tissue index; HbA1c—hemoglobin A1c; HDL—high-density lipoprotein cholesterol; LDL—low-density lipoprotein cholesterol; TyG—triglyceride-glucose index.

**Table 3 biomolecules-13-00697-t003:** Comparison of galectin-9 concentration.

Parameter	GDM	N	Mean	SD	Median	*p*
Galectin-9 in serum before delivery (ng/mL)	Absent	23	7.80	2.23	7.12	*p* = 0.074
	Occurring	18	9.39	3.02	8.80	
Galectin-9 in serum after delivery (ng/mL)	Absent	23	7.74	3.50	6.66	*p* = 0.253
	Occurring	18	8.40	3.09	7.52	
Galectin-9 in urine after delivery (ng/mL)	Absent	23	72.59	58.91	65.58	*p* = 1
	Occurring	18	78.31	71.77	72.19	

*p*—Mann–Whitney test; SD—standard deviation.

**Table 4 biomolecules-13-00697-t004:** Correlations with galectin-9 concentrations.

Parameter	Galectin-9 in Serum before Delivery	Galectin-9 in Serum after Delivery	Galectin-9 in Urine after Delivery
HbA1c (IFCC)	r = −0.064. *p* = 0.692	r = −0.179. *p* = 0.264	r = 0.002. *p* = 0.992
HOMA-IR	r = 0.298. *p* = 0.059	r = 0.13. *p* = 0.419	r = 0.043. *p* = 0.787
TyG	r = 0.161. *p* = 0.314	r = 0.194. *p* = 0.225	r = 0.137. *p* = 0.393
Total cholesterol	r = −0.037. *p* = 0.819	r = 0.157. *p* = 0.327	r = −0.076. *p* = 0.636
HDL	r = −0.217. *p* = 0.174	r = 0.06. *p* = 0.711	r = −0.126. *p* = 0.434
LDL	r = −0.032. *p* = 0.841	r = 0.117. *p* = 0.468	r = −0.215. *p* = 0.176
Non-HDL	r = 0.003. *p* = 0.984	r = 0.111. *p* = 0.489	r = −0.08. *p* = 0.618
Triglycerides	r = 0.06. *p* = 0.71	r = 0.173. *p* = 0.279	r = 0.128. *p* = 0.426
Homocysteine	r = 0.165. *p* = 0.304	r = 0.187. *p* = 0.241	r = −0.169. *p* = 0.292
BMI	r = 0.311. *p* = 0.048 *	r = 0.082. *p* = 0.612	r = 0.164. *p* = 0.304
E/I	r = 0.447. *p* = 0.003 *	r = 0.203. *p* = 0.202	r = −0.154. *p* = 0.335
ATM	r = 0.498. *p* = 0.001 *	r = 0.19. *p* = 0.234	r = 0.047. *p* = 0.772
FTI	r = 0.47. *p* = 0.002 *	r = 0.174. *p* = 0.276	r = 0.099. *p* = 0.537
BCM	r = −0.201. *p* = 0.208	r = −0.024. *p* = 0.88	r = 0.217. *p* = 0.173

*—Difference statistically significant (*p* < 0.05). ATM—adipose tissue mass; BCM—body cell mass; BMI—body mass index; E/I—extracellular water to intracellular water index; FTI—fat tissue index; HbA1c—hemoglobin A1c; HDL—high-density lipoprotein cholesterol; HOMA-IR—homeostasis model assessment–insulin resistance; LDL—low-density lipoprotein cholesterol; non-HDL—non-high-density lipoprotein cholesterol; TyG—triglyceride-glucose index.

**Table 5 biomolecules-13-00697-t005:** Galectin-9 in serum before delivery.

Parameter	Parameter	95%CI		*p*
HbA1c (IFCC)	−0.104	−0.317	0.11	0.351
HOMA	0.013	−0.046	0.072	0.671
TyG	1.922	−5.474	9.319	0.615
Total cholesterol	0.007	−0.052	0.065	0.819
HDL	0.001	−0.097	0.099	0.991
LDL	−0.01	−0.052	0.032	0.642
Triglycerides	−0.011	−0.055	0.032	0.619
Homocysteine	0.233	−0.126	0.592	0.214
BMI	0.003	−0.283	0.288	0.985
E/I	1.84	−8.018	11.697	0.717
ATM	0.059	−0.102	0.22	0.479
FTI	−0.067	−0.527	0.392	0.776
BCM	−0.015	−0.244	0.214	0.9

*p*—Multiple linear regression. ATM—adipose tissue mass; BCM—body cell mass; BMI—body mass index; E/I—extracellular water to intracellular water index; FTI—fat tissue index; HbA1c—hemoglobin A1c; HDL—high-density lipoprotein cholesterol; HOMA-IR—homeostasis model assessment-insulin resistance; LDL—low-density lipoprotein cholesterol; non-HDL—non-high-density lipoprotein cholesterol; TyG—triglyceride-glucose index.

**Table 6 biomolecules-13-00697-t006:** Galectin-9 in serum after delivery.

Parameter	Parameter	95%CI		*p*
HbA1c (IFCC)	−0.095	−0.378	0.189	0.518
HOMA	0.021	−0.057	0.098	0.604
TyG	3.562	−6.232	13.356	0.482
Total cholesterol	−0.041	−0.118	0.037	0.314
HDL	0.082	−0.048	0.211	0.228
LDL	0.025	−0.03	0.081	0.38
Triglycerides	0.008	−0.05	0.066	0.79
Homocysteine	0.244	−0.231	0.719	0.323
BMI	−0.011	−0.389	0.367	0.955
E/I	6.654	−6.4	19.707	0.327
ATM	0.043	−0.17	0.256	0.696
FTI	−0.183	−0.791	0.426	0.561
BCM	0.096	−0.207	0.399	0.134

*p*—Multiple linear regression. ATM—adipose tissue mass; BCM—body cell mass; BMI—body mass index; E/I—extracellular water to intracellular water index; FTI—fat tissue index; HbA1c—hemoglobin A1c; HDL—high-density lipoprotein cholesterol; HOMA-IR—homeostasis model assessment-insulin resistance; LDL—low-density lipoprotein cholesterol; non-HDL—non-high-density lipoprotein cholesterol; TyG—triglyceride-glucose index.

**Table 7 biomolecules-13-00697-t007:** Galectin-9 in urine after delivery.

Parameter	Parameter	95%CI		*p*
HbA1c (IFCC)	−1.274	−6.781	4.233	0.654
HOMA	0.397	−1.113	1.907	0.611
TyG	−0.997	−191.449	189.454	0.992
Total cholesterol	−1.188	−2.695	0.319	0.134
HDL	1.672	−0.851	4.195	0.205
LDL	0.392	−0.693	1.477	0.485
Triglycerides	0.52	−0.602	1.642	0.372
Homocysteine	1.397	−7.844	10.637	0.769
BMI	3.474	−3.874	10.821	0.362
E/I	−32.736	−286.576	221.105	0.802
ATM	−3.031	−7.179	1.116	0.163
FTI	6.123	−5.707	17.953	0.319
BCM	0.895	−4.997	6.787	0.768

*p*—Multiple linear regression. ATM—adipose tissue mass; BCM—body cell mass; BMI—body mass index; E/I—extracellular water to intracellular water index; FTI—fat tissue index; HbA1c—hemoglobin A1c; HDL—high-density lipoprotein cholesterol; HOMA-IR—homeostasis model assessment-insulin resistance; LDL—low-density lipoprotein cholesterol; non-HDL—non-high-density lipoprotein cholesterol; TyG—triglyceride-glucose index.

## Data Availability

Not applicable.

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
