# Peer review of "Do Serum Galectin-9 Levels in Women with Gestational Diabetes and Healthy Ones Differ before or after Delivery? A Pilot Study"

_biomolecules, 2023, doi:10.3390/biom13040697_

Round 1

Reviewer 1 Report

Revised manuscript undoubtedly has strong potential due to novelty of described issues as well because of incidence of gestational diabetes mellitus. However in presented form manuscript cannot be published.

In general:

1. Reformulation of whole manuscript because results does not support conclusions.

2. Rationale of the study is not clearly presented.

3. Plenty of the results, most of them does not deserve presentation in the manuscript, whereas the reader has problems in focusing on those which are crucial in perspective of hypothesis (missing).

4. References in general are ok, however there are some positions which should be introduced.

5. Tiny study groups. In such perspective presented study should be treated as a preliminary/initial study. 

6. Concussions does not adhere to the results… (results may be associated with obesity, but there is no relation with gdm…)

In detail

1. Do we know the exact excretion profile of galectin 9? Why urine was obtained by one probe not in long time period together? what is the T1/2 for galectin 9? Collection of the samples during whole hospitalizations?

2. In my opinion inclusion and exclusion criteria are insufficiently described. What about cl. 1 and cl. 2? Once again, only 18 GDM patients; it is not rare complication…

3. What was the rationale for testing in early postpartum period especially if the clearance rate was not pointed out?

4. What was the rationale for presentation so many data in e.g. table 2. What was the distribution of parameters, acc. to this knowledge we present ‘mean’ or ‘median’ and accessory…

5. Figures showing correlations have no superiority over a plain description in the text.

6. In conclusions, which should be reformulated due to pointed out reasons above, the authors cannot duplicate results. 

… Moreover, in regard to achieved results /table 2/ we may hypothesize that if galectin 9 correlates positively with markers ‘associated’ with adipose tissue in GDM patients, subsequently gestation eliminates correlations, but we still don’t know how fast we should expect clearance of galectins, so it’s clear that even such speculation /much more justified than presented by the authors/ is not convincing, and does not correspond to the aim of the study… 

Reviewer 2 Report

Manuscript ID: Biomolecules-2323057

The authors evaluated Galectin-9 expression in pregnancies complicated by GDM. As the authors wrote, there are no publications exploring this topic.

Without a doubt, research on Galectin biomarkers is very interesting and can contribute to the understanding how GDM influences early onset of future complications.

The authors results showed that there are no significance differences in Galectin-9 levels between normal pregnancies and GDM in serum and urine samples.

The subject of the research is interesting and it is necessary to deepen and investigate it. Despite this, I am concerned about the women's data which were presented in the manuscript and on which the entire analysis was based. It is necessary to provide more details to Table 1 in order to ensure that the women were actually diagnosed with GDM.

1.      The BMI values were not significantly different between the GDM women and normal pregnancy. It is well-known the BMI of GDM women are significantly higher than those with normal pregnancies. How do the authors explain these results?

2.      The glucose levels presented in Table 1 – even though the authors found significant differences between the 2 groups, the levels presented are in the normal range even for GDM women.  How do the authors explain this?

3.      The authors need to add to the Methods the screening protocol used to diagnose the pregnant women with diabetes. 

4.      The authors need to add to Table 1 the first trimester fasting blood glucose levels of women as well the glucose levels measured during the glucose challenge test (GCT) and oral glucose tolerance test (OGTT).  

5.      Please provide information in  Table 1 whether the GDM patients received therapy during the course of pregnancy.

6.      The title should reflect the main finding of the manuscript , please edit it.

7.      The authors should expand the Introduction and include recent studies of other galectins, such as galectin-3 and gestational diabetes mellitus.

8.      Expanding the research of Galectin-9 expression in placental tissue as well as fetal (blood cord) will be very interesting.

Reviewer 3 Report

In the manuscript "Galectin-9 in gestational diabetes mellitus" the authors compared galectin-9 levels in healthy pregnant women and those with GDM. This is the first publication regarding galectin-9 involvement in GDM and the authors very thoroughly analysed obtained results. I only have few comments.

-          Please comment whether the diabetic patients were on a special diet or received any therapy.

-          It would be good to know in which week of pregnancy the patients were diagnosed with GDM, and in a subsequent study to monitor the risk group in the context of galectin-9 from the beginning of pregnancy.

-          There are two modes of delivery in both investigated groups. Is there any difference between measured galectin-9 within groups?

-          Enninga et al. (2018) shows significantly higher galectin-9 levels in woman carrying male fetuses than those carrying females. Did the authors observe such a difference in this study?

-          Did the authors measure creatinine in urine?

Round 2

Reviewer 1 Report

I see significant improvement of revised manuscript. The Authors introduced all suggested amendments. In present form I suggest acceptance for publication. 

Reviewer 2 Report

The revised manuscript has improved significantly as a results of the authors positively response to the reviewers comments.

The only comment I have about the new version is that Figs 1 and 2 can be removed since all data are well explained in the Methods section.

please check Ref 39

Reviewer 3 Report

In the revised manuscrpit, the authors answered my concerns. I only have minor comments:

 -          Last sentence in introrduction – is galectin-9 missing after urine….as a marker of GDM?

-          Discussion – second paragraph. Meanwhile, after delivery – maybe it is better after labor?
